# Improved Interpolation and Anomaly Detection for Personal PM$_{2.5}$ Measurement

**JinSoo Park [1] and Sungroul Kim [2,*]** 

[1] Department of Industrial Cooperation, Soonchunhyang University, Asan 31538, Korea; vtjinsoo@gmail.com
[2] Department of Environmental Sciences, Soonchunhyang University, Asan 31538, Korea
[*] Correspondence: sungroul.kim@gmail.com; Tel.: +82-41-530-1266

**Abstract:** With the development of technology, especially technologies related to artificial intelligence (AI), the fine-dust data acquired by various personal monitoring devices is of great value as training data for predicting future fine-dust concentrations and innovatively alerting people of potential danger. However, most of the fine-dust data obtained from those devices include either missing or abnormal data caused by various factors such as sensor malfunction, transmission errors, or storage errors. This paper presents methods to interpolate the missing data and detect anomalies in PM$_{2.5}$ time-series data. We validated the performance of our method by comparing ours to well-known existing methods using our personal PM$_{2.5}$ monitoring data. Our results showed that the proposed interpolation method achieves more than 25% improved results in root mean square error (RMSE) than do most existing methods, and the proposed anomaly detection method achieves fairly accurate results even for the case of the highly capricious fine-dust data. These proposed methods are expected to contribute greatly to improving the reliability of data.

**Keywords:** data interpolation; anomaly detection; bootstrap; fine dust; PM$_{2.5}$

## 1. Introduction

Korea has been experiencing severe environmental health problems caused by exposure to fine dust [1]. Thus, many stakeholders, including government officials, are trying hard to find solutions for the environmental issues. As part of these efforts, various artificial intelligence (AI)-based technologies are drawing attention as a way to predict future exposure levels as well as to reduce real-time exposure to PM$_{2.5}$ in our daily life. PM data is closely related to personalized health-care service and preventive medicine, which are research areas that have attracted much interest from many researchers today. The personalized healthcare service prompted us to develop predictive analytics technology, which requires the acquisition of data related to individual activity patterns [2,3]. Such data can be seen as person-specific data that is different from the population-based data to be used for the existing broadcasting-type environmental information service aimed at a large audience [2,4]. The pico-scale data is usually collected from each individual sensor device [4]. Unfortunately, such data are more likely to be incomplete than data collected from stationary sensors, because sensors attached to human subjects are affected significantly by the person's activity patterns, meteorological conditions, or the malfunction of the installed device or sensor itself. These kinds of incomplete data can lead to the provision of wrong services, because they may invoke wrong algorithmic decisions caused by the data. Such incomplete data typically include missing or abnormal data. In order to provide high-quality environmental information services, it is essential to conduct studies to deal with these two issues.

Much research has been done on the two issues mentioned above. Most of such research tends to focus on detecting missing or anomalous data only as follows. Conventional techniques related to missing data imputation in time-series data include methods based on machine learning, such

as random forests [5], maximum likelihood estimation [6], expectation maximization [7], or nearest neighbors [8]. Technologies belonging to anomalous data detection include prediction-based [9], distance-based [10], probability-based [11], and linear models [11]. Researches dealing with both technologies include [12,13] both using machine learning techniques. As we have seen, there are few studies that simultaneously address these two problems.

Thus, in this study, we present our data-mining skills approach that deals with the two problems of missing and abnormal data included in the PM$_{2.5}$ data obtained from a personal portable sensor. Especially, we demonstrate that our kernel regression-based interpolation method and abnormal data detection method can be applied to our real personal PM$_{2.5}$ measurement data. We attempt to extend a well-known interpolation method incorporating a simple linear interpolation method to interpolate the bursty PM$_{2.5}$ data. The performance of the proposed method is provided in comparison to those of existing interpolation methods. In addition, details on the method are presented in the algorithmic method section of this paper.

## 2. Methods

### 2.1. Proposed Algorithm

In this paper, we proposed two algorithms: the kernel regression-based interpolation method and the subsequent abnormal-data detection method. The algorithm presented in this paper was done in the order shown in Figure 1. First, we chose the part of the total data that had no missing data and estimated the bandwidth for the chosen data part. Here, the bandwidth was the value to use for the kernel regression-based interpolation (KRBI) (dotted line in Figure 1), which will be explained in the next subsection. Afterwards, we examined missing data for the entire dataset. If there was missing data, we used linear interpolation (LI) to interpolate the missing data. Afterwards, the interpolation was done again by applying the KRBI algorithm [14]. In this case, the optimal bandwidth value, which was previously obtained, was used. If there were no missing data, abnormal-data detection began for the entire dataset.

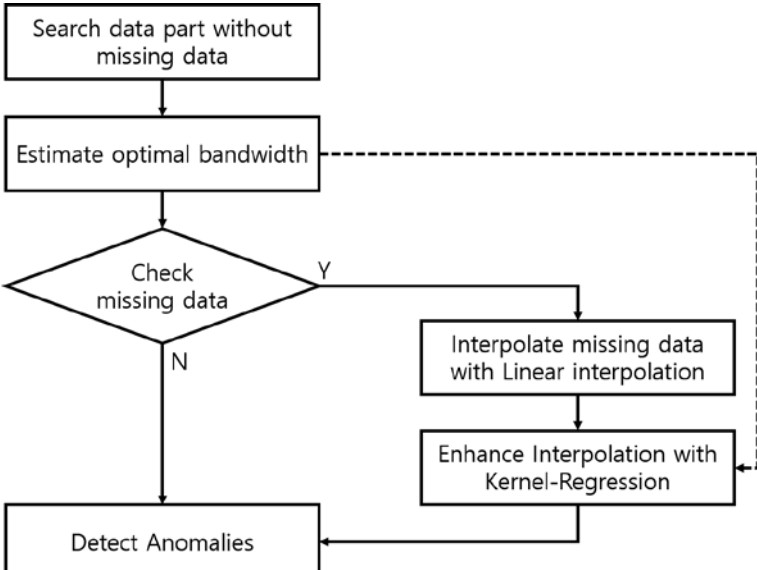

**Figure 1.** Flowchart of the proposed interpolation and anomaly detection.

### 2.2. Optimal Bandwidth Selection Based on Leave One Out Cross-Validation (LOOCV)

In order to calculate the appropriate bandwidth, we took all of the data and split it into training and verification data. As the bandwidth was changed from the small value to the large value, the verification data value was predicted using the training data. Finally, we calculated the estimated

error in terms of the actual verification data value. Among the bandwidth values that contributed to the error calculation, we programmed our algorithm to take the bandwidth value that provided the lowest error as an appropriate bandwidth for the data used. The corresponding pseudo-code is available in [15]. The bandwidth selection process is explained in the experimental section.

### 2.3. Kernel Regression-Based Interpolation Using Linear Interpolation

The method proposed in this paper is based on the kernel regression [14], but it took the advantages of the linear interpolation method [16]. The LI method can be used appropriately, especially when the time-series trend is clear. For instance, when the time-series pattern appears to be rising up or decreasing, the LI method can be applied for the interpolation of the corresponding data pattern more appropriately. We used this property of the LI method in order to improve the performance of the KRBI method. In other words, we used the LI method for the bursty missing data a priori and then applied the KRBI method for the final interpolation of the missing data.

The kernel regression algorithm can be summarized as follows. First, we defined the time-series data as $(t_i, y_i)$ where $t_i$ and $y_i$ represented the time and measurement of data at time $t_i$. Kernel regression was to set the representative value $\hat{y}$ of $y_i$s where $p \leq i \leq q$ and the bandwidth $h$ are defined as $h = p - q$. In this case, the representative value $\hat{y}$ might be calculated as a weighted average value of $W_i y_i$ where the weight $W_i$ could be generated by following well-known statistical models such as Gaussian or uniform distributions. The algorithm can be expressed mathematically as follows.

$$\hat{y} = \frac{\sum_{i=1}^{n} K_h\left(\frac{x-x_i}{h}\right)y_i}{\sum_{j=1}^{n} K_h\left(\frac{x-x_j}{h}\right)} = W_i y_i$$

where the choice of weight $W_i = \frac{\sum_{i=1}^{n} K_h\left(\frac{x-x_i}{h}\right)}{\sum_{j=1}^{n} K_h\left(\frac{x-x_j}{h}\right)}$ and $K_h(\cdot)$ is a kernel chosen [14].

In order to apply the KRBI method, a proper bandwidth calculation must be done. For this purpose, we used a part of the data that had no missing data to estimate the proper bandwidth for the data interpolation. A detailed description is provided in the next section.

### 2.4. Context-Aware Anomaly Detection

Once we finished the bandwidth selection for interpolation and optimal bandwidth, we developed another algorithm for detecting anomalies in the time-series data. Many techniques have been presented to detect anomalies using various techniques. In this paper, we defined anomalies as data that showed significant changes in values within a very short time. For instance, if $PM_{2.5}$ concentration, measured every 10 seconds, showed significant drops or jumps during the 10-s period (for instance, observations that fall below $Q_1$-1.5IQR or above $Q_3$+1.5IQR in the box-and-whisker plot), we considered the value to be abnormal, because the amount of change in $PM_{2.5}$ concentration is assumed to stay stable or similar within a very short time. However, this rule was not an appropriate criterion for detecting outliers, because too much data fell into this category, and it was hard to think that they were all anomalies, given the data context. Thus, in this research, we defined data as anomalous when the following conditions were met. In reality, the $PM_{2.5}$ concentration does not change significantly in most cases. This phenomenon is reflected in the detection of anomalies in our time-series data analysis.

$$d_i = \left|y_i - y_{i-1}\right| > th$$

In other words, if $d_i$ (difference of adjacent $y_i$s) exceeds a certain threshold, *th*, then $y_i$ can be an anomaly. Thresholds are chosen by visual inspection according to the characteristics of the PM data at the moment. Details are available in the experimental section.

## 3. Experimental Tests

### 3.1. Bootstrap Simulation on Real Dataset

In this section, we verified the effectiveness of the proposed interpolation method. In order to verify its validity, we (1) randomly removed some of the actual data, (2) interpolated the removed data based on our algorithm, and then (3) compared them with each other in terms of certain performance criteria, including a comparison of applying results using other known methods. We executed our experiments based on a bootstrapping test using the given data in three different scenarios. We assumed that the arc shape of time-series data could be classified into three different patterns in general: up slope, down slope, and flat. Based on this assumption, the validity of the method could be evaluated only for data belonging to each shape pattern. We used the bootstrapping test on the chosen dataset belonging to each shape pattern. Samples from each pattern section of the data were randomly selected and then deleted on purpose. Next, we estimated the interpolating values for the deleted data and compared the error rate between the real values and the estimated interpolation values. The time-series data for these validation tasks are given in Section 3.3.2. Given our thorough examination of the data, the data corresponding to the following time index were specifically chosen because they appeared to match the three typical shape patterns and did not have missing values in the patterns: 3600 to 4600 for the up slope, 3250 to 3500 for the down slope, and 19,000 to 20,000 (Flat 1), 27,000 to 28,000 (Flat 2), and 37,000 to 40,000 (Flat 3) for the flat pattern. Specifically, for the flat pattern case, we chose three sections in order to examine any significant performance differences, because the duration of the flat pattern section is relatively longer than the other part of the time-series data. In our bootstrapping test, 40, 60, 80, and 100 datapoints were randomly deleted to create missing data, and the interpolation results on the deleted data were evaluated compared to those of the original data in terms of RMSE (root mean squared error).

The performance of our proposed method was compared to those of the LOCF (last observation carried forward), Agg (aggregate), and Spline methods [17]. The LOCF method takes the most recent values prior to itself. The Agg method takes the mean of a few previous values. The Spline method is a smoothing technique that comes with base R; it was used for the interpolation of missing data. The results of comparing these four methods are given in Table 1, which shows that our proposed method had lower RMSEs than those of the existing three methods, except for the flat pattern, for which the RMSEs of all four methods were not statistically different. R programming language was used to analyze the performance of the interpolation methods, and R packages stats, zoo, and Metrics were used for the interpolation and calculation of error rates [18]. The simulation was performed on a Microsoft Windows 10 computer with Intel® Core™ i7-6500U CPU at 2.60GHz.

As demonstrated by the real data-set experiments, our proposed method worked better than the existing methods did, although there were very similar results for a few cases, such as the flat pattern. Based on these experimental results, we finally applied our proposed methods to the remaining cases of missing values.

**Table 1.** Root mean squared errors (RMSEs) of the four interpolation methods applied to our real-time personal monitoring data. LOCF: last observation carried forward.

| Data Pattern | Interpolation Method | Number of Missing Data | | | |
|---|---|---|---|---|---|
| | | 40 | 60 | 80 | 100 |
| Up slope | Proposed | 26.919 | 26.426 | 25.733 | 28.599 |
| | Spline | 37.052 | 36.679 | 35.682 | 37.908 |
| | LOCF | 36.604 | 36.909 | 35.707 | 38.393 |
| | Agg | 656.741 | 657.476 | 659.759 | 658.761 |
| Down slope | Proposed | 24.687 | 25.704 | 27.840 | 28.945 |
| | Spline | 26.741 | 27.817 | 30.233 | 31.834 |
| | LOCF | 34.223 | 35.888 | 38.719 | 39.818 |
| | Agg | 283.824 | 278.758 | 280.763 | 280.666 |
| Flat 1 | Proposed | 4.551 | 3.981 | 4.341 | 4.376 |
| | Spline | 3.871 | 3.326 | 3.555 | 3.535 |
| | LOCF | 5.215 | 4.803 | 4.874 | 5.403 |
| | Agg | 4.426 | 3.918 | 4.052 | 4.071 |
| Flat 2 | Proposed | 3.633 | 3.751 | 3.694 | 3.656 |
| | Spline | 3.505 | 3.610 | 3.514 | 3.493 |
| | LOCF | 4.554 | 4.817 | 4.893 | 4.783 |
| | Agg | 4.031 | 4.129 | 4.062 | 4.062 |
| Flat 3 | Proposed | 2.335 | 2.310 | 2.429 | 2.325 |
| | Spline | 2.229 | 2.080 | 2.237 | 2.187 |
| | LOCF | 2.991 | 3.027 | 3.226 | 2.875 |
| | Agg | 2.271 | 2.125 | 2.293 | 2.233 |

### 3.2. Optimal Bandwidth Selection

Before we carried out our interpolation on the data, we needed to decide on the value of the bandwidth used for our interpolation method. For this task, we randomly chose a complete part of the data that did not have missing data to estimate optimal bandwidth for the data set. Then, the complete data were split into two different sets for training and validation. The training dataset was used to set up a kernel regression model, and then the validation was carried out for the remaining data set. In this bandwidth-selection step, we increased the bandwidth from 1 to 100 to find out which bandwidth provided the smallest error that could be applied to our interpolation procedure. This bandwidth estimation step can in fact depend on the nature of the data. That is, since the results of the estimation may differ significantly depending on the shape of the data pattern, different bandwidths for each of the three data patterns were estimated and used for this study. However, because it was hard to detect the changing point of these different data patterns in our time-series data, we used the bandwidth estimated for the flat data pattern for the following two real-world data experiments. Further research on automatically calculating the appropriate bandwidth depending on the data patterns is necessary in the future.

### 3.3. Interpolation and Anomaly Detection with Real-World Personal Data

We applied our method described above our separated $PM_{2.5}$ data set containing missing data. These $PM_{2.5}$ data were collected at 10-s intervals from portable personal $PM_{2.5}$ monitors attached to the subjects. The data were measured between 25 January 2019 and 1 February 2019.

3.3.1. Application of Interpolation and Anomaly Detection Method with Real-World Personal Dataset 1

As shown in Figure 2, the level of $PM_{2.5}$ was stable, implying that the subject was relatively calm with minimal abrupt activity changes. The length of the dataset was 59,422 at 10-s intervals, but it

had 17,968 missing datapoints in total. Before we went on to detect the anomalies, we first selected an optimal bandwidth for the dataset as we did previously.

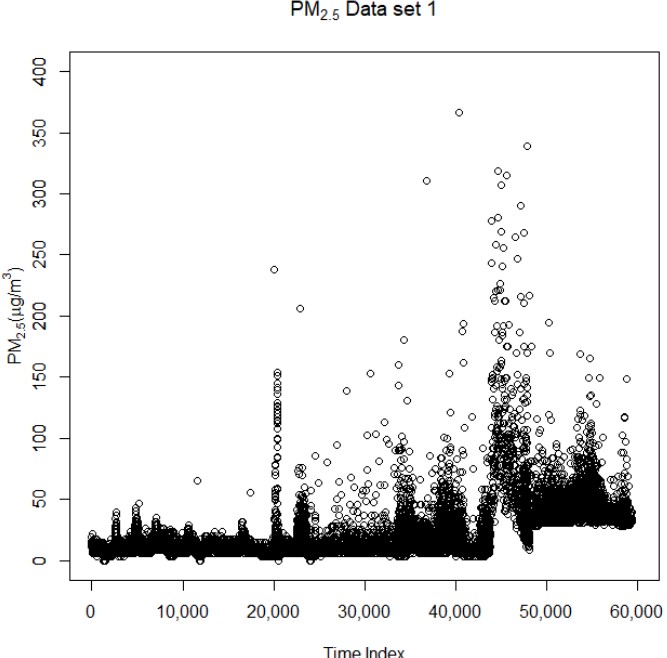

**Figure 2.** This is a data set that was collected for 6 days between 25 January 2019 and 31 January 2019. The length of the dataset is 59,422, having 17,968 missing datapoints in total.

After deciding on an optimal bandwidth for the data set, as seen in Figure 3, we interpolated the missing values based on the algorithm mentioned before. The corresponding results are given in Figure 4.

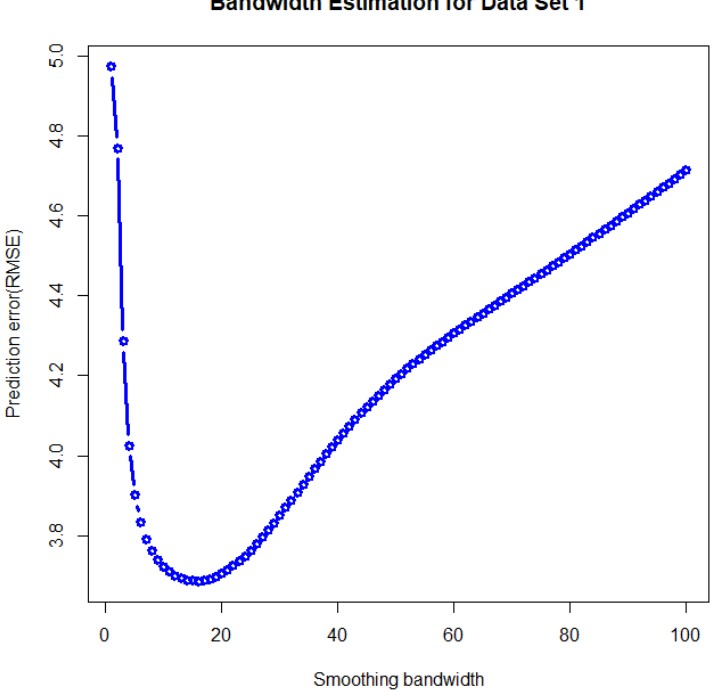

**Figure 3.** By increasing the bandwidth from small to large, we calculate the RMSE of the estimation and examined at what value of the bandwidth the corresponding RMSE is the minimum.

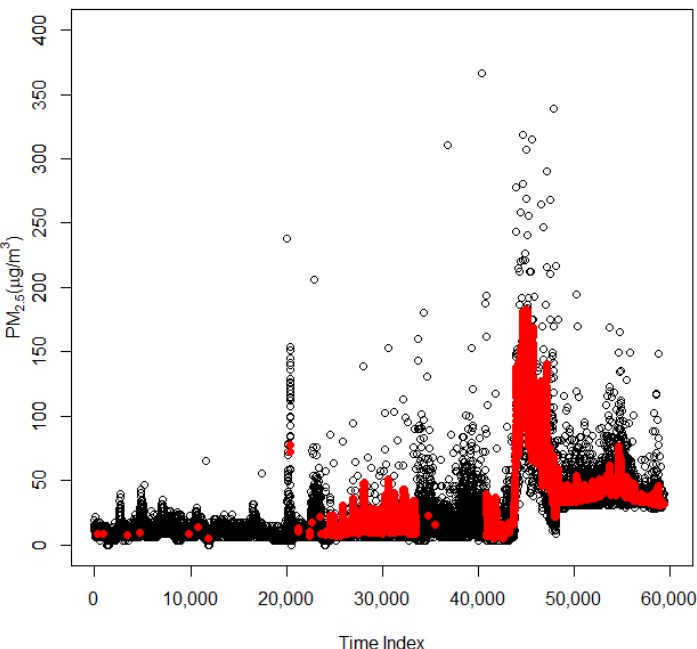

**Figure 4.** Missing values are interpolated and shown in red.

Figure 4 shows the interpolated data (red dots) superimposed on the actual data. It indicates that the missing data appeared in a small or large aggregated formation. In particular, it can be seen that much of data loss occurred between 40,000 and 60,000 according to the time index. After checking the raw data, we confirmed that this large data loss occurred between 13:00 and 16:00, presumably because of various activities in the afternoon. So much missing data could be very difficult to interpolate by methods other than linear interpolation. In such a case, it is preferable to carry out the initial interpolation using linear interpolation, as in our method, and then to apply other methods to improve the result. As shown in the figure, the interpolation goes well with the overall pattern of the data.

As the next step, for the entire dataset, we detected anomalies in the dataset, based on the method described in the method section. When the difference of adjacent $PM_{2.5}$ values is above a certain threshold (in this experiment, 200), we considered them to be anomalies. The corresponding anomalies are shown in red in Figure 5, which shows there are eight anomalies detected by visual inspection; the four red dots on the top seems to be the true outliers. The other red dots at the bottom do not appear to be true outliers, but they can also be regarded as outliers, because the PM concentration significantly dropped from the previous state by more than 200 $\mu g/m^3$ in 10 s, which is not acceptable as a normal degree of change. This result explains that the proposed outlier detection method produces fairly reliable results to some degree, even in a highly capricious environment. In selecting a threshold, we took an empirical approach to decide the threshold. Some data that could be visually regarded as anomaly data were selected as reference data, and then these data were examined to see if they were detected as actual anomaly data. As we performed the experiments with varying thresholds, the values when these reference data were detected were selected as thresholds.

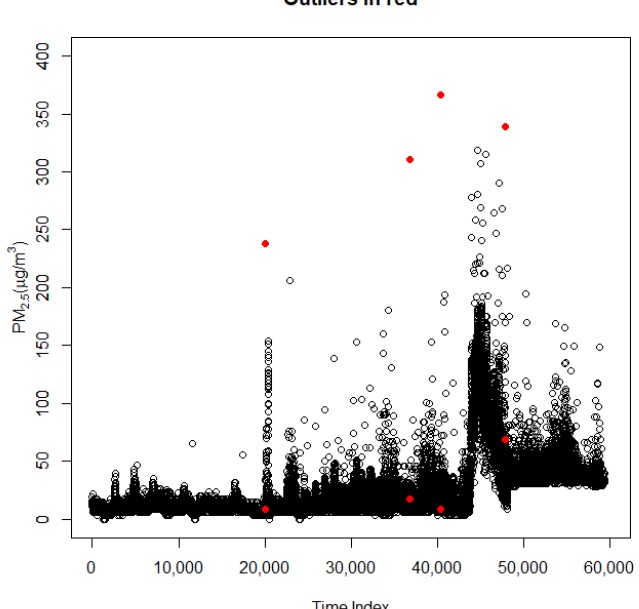

**Figure 5.** Anomalies are shown in red.

### 3.3.2. Application of Interpolation and Anomaly Detection with Real-World Personal Dataset Two

This dataset was collected for 8 days between 25 January 2019 and 1 February 2019. The length of the dataset is 62,878, having 74 missing datapoints in total. Dataset 1 used in the previous experimental test was very stable, because the distribution of PM data was mostly less than 100 $\mu g/m^3$ during the data acquisition period. However, the PM data in Dataset 2 (Figure 6) reached up to 2000 to 8000 $\mu g/m^3$ with 10-s intervals and showed a more dynamic change in the distribution of PM data, implying that the subject had various activities or was exposed to many different environmental conditions containing all the data-distribution patterns of rising, falling, and stable $PM_{2.5}$ concentrations.

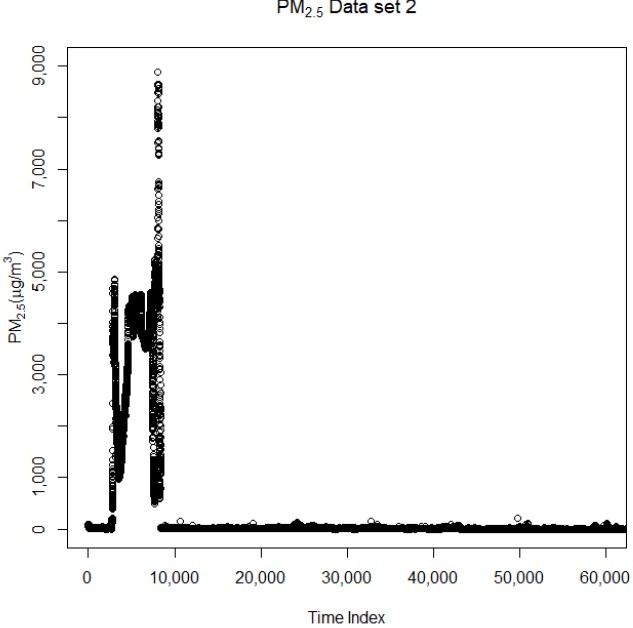

**Figure 6.** This is Dataset 2, collected for 8 days between 25 January 2019 and 1 February 2019. The length of the dataset is 62,878, having 74 missing datapoints in total.

The same bandwidth estimation steps were also carried out as described previously. Figure 7 shows the chosen bandwidth of 20 estimated for the flat part of the data between 33,000 and 33,600 in the time index. The interpolated missing data are shown in red in Figure 8, overlaid on the entire dataset.

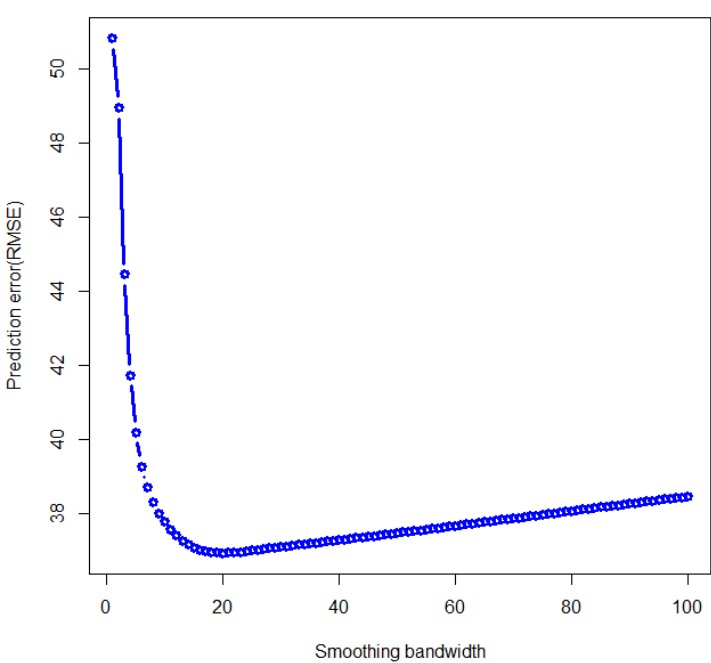

**Figure 7.** Optimal bandwidth for this dataset is chosen as being 19.

**Interpolated Data**

**Figure 8.** Missing data is interpolated and shown in red.

Figure 8 can be seen as two sets of data with different natures; in the front part, the data fluctuates heavily, but in the back part, the data are relatively stable. In particular, the front data were generated between 14:00 and 18:00, probably by a wide variety of movements. Unlike the previous Dataset 1,

the data loss is relatively small, even under the condition of dynamic movements; so, a very stable performance sensor may be used. In the case of the latter part of the data, it appears that more data loss occurs than from the front part, but not much data are actually lost. Overall, a fairly stable interpolation result can be observed.

Finally, anomalies are also shown in red, in Figure 9. Overall, eight anomalies seem to be detected by visual inspection. If you look at the enlarged part of the data (between 2750 and 4000 out of the whole dataset), you can see that the detected outliers can be accepted as real outliers, because the corresponding PM concentration significantly changes within a 10-s period. In this experiment set, the threshold value 1000 was used as the criterion to detect outliers, and the threshold was also empirically chosen, analogously to the approach described in Section 3.3.1.

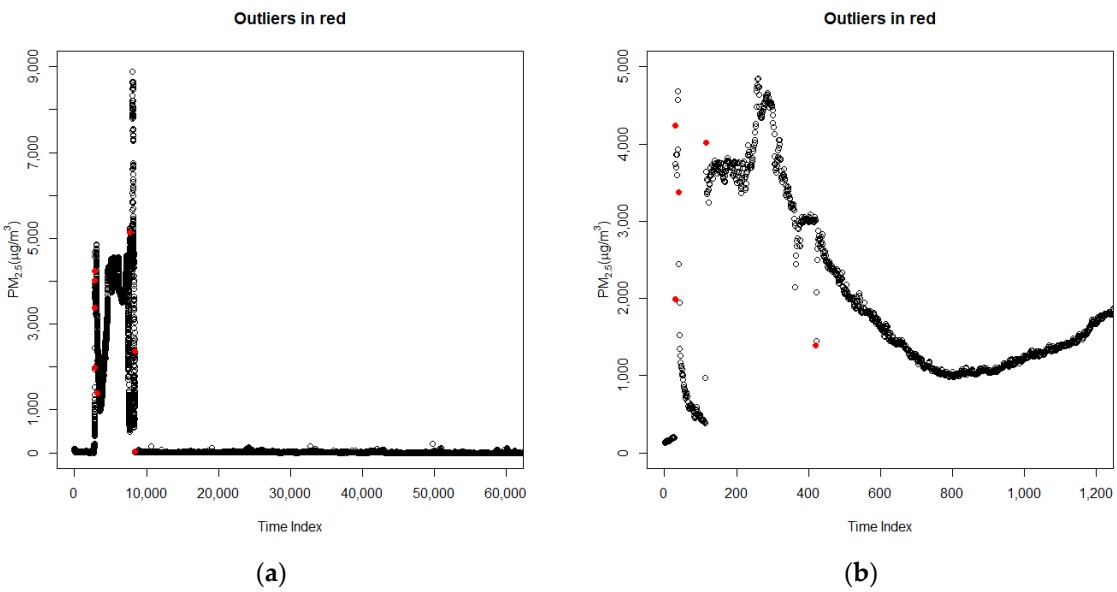

(a)                                    (b)

**Figure 9.** Anomalies are detected and shown in red (**a**) outliers for the entire dataset (**b**) outliers shown enlarged for the part of the dataset between 2750 and 4000.

## 4. Discussion

This paper presented the results of research on two important technologies related to artificial intelligence environmental information services, such as $PM_{2.5}$ exposure level prediction, and providing alerts based on the prediction. Most environmental data inevitably contain both missing data and anomalies caused by various factors, such as sensor malfunctions, errors in transmission, or errors in storage. Such incomplete data can lead to the miscalculation of data-based analysis and so must be processed appropriately before we use them for data analysis. Most of the related studies have tended to deal with either interpolation or abnormal data problems only. However, it is desirable that interpolation and abnormal data should be handled together for viable data-based services. Therefore, we conducted the research on these two problems simultaneously.

The first technique was interpolation, which can be regarded as a technique to cope with data lost from various sensors. Data tend to be missed mainly in time-series data, which is known to occur in three major forms: missing completely at random (MCAR), missing at random (MAR), and not missing at random (NMAR). In particular, the MAR type is an appropriate form of model for describing the data that are missing in most fine-dust time-series data [17]. This model assumes that the pattern of the time-series data can be described by a certain mathematical generative model and also attempts to take advantages of correlation information in multivariate environments to solve the missing-data problem [17]. In a stationary environment where sensors are fixed to certain objects, the generative model may be used to analyze the data. However, in a univariate case where data are collected from portable wearable devices, it is very challenging to estimate the missing data, because the number of

attributes to use is very limited, and a large amount of data is likely to be missing [19]. In these cases, it is very difficult to apply a mathematical generation model to describe the data-distribution pattern or representative values for a specific activity. There are very few articles addressing the interpolation methods for univariate time-series data. Articles by Junninen studied the univariate algorithm in 2004, but do not consider time-series aspects [20]. Authors applied ARIMA (autoregressive integrated moving average) and SARIMA (seasonal autoregressive integrated moving average) models for the univariate model interpolation and provided comparison results [21]. A performance comparison is provided, using built-in interpolation methods in R [19]. Data loss in one instant or a short period of time can be easily interpolated with a simple error-recovery method such as linear interpolation. However, this simple interpolation method may be inadequate for a long bursty loss of sensor-provided data. When such long bursty data loss occurs, the interpolation method with the prediction technique may be more appropriate.

In addition, anomalous-data detection technology should be applied to detect data that deviate from the characteristics of data distribution. The anomalies can be commonly found in various industries including the environmental and finance fields. Anomalous time-series data can be regarded as data that disturb the continuity of data based on temporal flow [11]. Most techniques to detect anomalies can be classified as supervised and unsupervised according to the presence or absence of data labels [22]. The supervised method is a technique for detecting abnormal data by means of a learning algorithm when the data are labeled. The unsupervised method can be used when there is no label in the data, and can be used more flexibly than the supervised method can be, because often data labels are not available. As another category of classification, point-anomalous data detection technology detects one datapoint that has abnormal characteristics among much normal data [22]. In contrast, the statistical method extends the point method, and is a technology that detects data when that fall inside or outside a specific range of values in order to find anomalies. The disadvantage of these technologies is that most of these values are set up by hand. Recently, a lot of context-based methods have been studied. These can be seen as techniques for identifying abnormal data depending on the context of a situation. We proposed a method to detect abnormal data that show significant incremental or decremental changes as measured by time.

Although the excellence of the proposed algorithms has been proved by experiments, we address the limitations in conducting this study. First, the bootstrapping test was used to prove the excellence of the interpolation method, because we did not have any reference data for our interpolation algorithm. However, under such circumstances, we assumed that the bootstrapping test was the best choice for generating a missing value dataset randomly from a secure dataset. In addition, we did three scenario-specific experiments to evaluate the performance of the proposed method, which could be considered a simplified approach. The patterns of data distribution can be more complicated; since data distribution patterns were not considered in this study, some other methods might work better, depending on the type of data. In addition, we did interpolation by applying the same bandwidth estimated for the flat data pattern to the entire dataset. In the future, it is necessary to conduct research to automatically apply different bandwidth values for each data pattern. Finally, because the method proposed in this study was a kind of context-aware based detection method, we acknowledge that other methods previously proposed could get a completely different result. In addition, the threshold value was chosen empirically by visual inspection at the moment, but we need to develop a sophisticated algorithm to choose the value automatically. We may introduce such an automatic method in our near future study.

Despite the limitations of this study, it has a very important academic significance in that it presents a solution to the problem of interpolation and abnormal data of fine-dust data acquired from personal mobile terminals. We believe that these technologies will be a cornerstone for personalized environmental data services in the near future. Future research will include the prediction of PM data concentrations in both indoor and outdoor environments based on machine-learning technologies.

## 5. Conclusions

We think out findings have contributed greatly to overcoming the incompleteness of environmental data obtained from individual sensors and to providing an academic basis for more reliable data analysis. If the proposed algorithm is further improved, it will contribute a lot to advancing personalized healthcare and preventive medicine research.

**Author Contributions:** Conceptualization, J.P. and S.K.; Methodology, J.P.; Software, J.P.; Validation, J.P. and S.K.; Data Analysis, J.P. and S.K.; Resources, J.P.; Data Curation, J.P. and S.K.; Writing—Review and Editing, J.P. and S.K.; Visualization, J.P.; Project Administration, S.K.; Funding Acquisition, S.K. All authors have read and agreed to the published version of the manuscript.

**Funding:** This study was funded by The Environmental Health Research Center Project (2016001360003) by The Korea Environmental Industry and Technology Institute, Ministry of Environment, South Korea. This research was also supported by the Soonchunhyang University Research Fund.

**Acknowledgments:** Authors thank the study participants and their parents.

**Conflicts of Interest:** The authors declare no conflict of interest.

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
