# Peer review of "Improved Interpolation and Anomaly Detection for Personal PM2.5 Measurement"

_applsci, doi:10.3390/app10020543_

Round 1
Reviewer 1 Report
The manuscript is of interest given the mounting usage of portable air quality devices. The manuscript is concise and generally well written, but I suggest to address this points for its improvement:
line 34: what is the exact definition of "pico-scale"? line 99: please define and explain what is "Kh" Figures 6-9 and related text: there extremely high values of PM2.5 shown here, up to >8000 ug/m3. These are barely credible as realistic values: did the author double checked the data, e.g. where and in which environment they were detected? I understand that this is mostly a paper presenting a new time series method, thus the physical meaning of the values may be overridden to some extent, but this seems to be really too much. I suggest to select another sample series to illustrate the results, e.g. the part after time 10.000 of the same series. For reproducibility of the work, I strongly suggest to document the software that was used. At line 149 is mentioned R, but a detailed documentation of the calculation environments and packages used should be included, with proper acknowledgement and references.
Author Response
To reviewer 1:
line 34: what is the exact definition of "pico-scale"?
Thank you for asking. To provide clarification, we changed the sentence containing “pico-scale” as seen below.
(After) The personalized healthcare service prompted us to develop predictive analytics technology, which requires the acquisition of data related to individual activity patterns [2,3]. Such data can be seen as person-specific data that is different from the population based data to be used for the existing broadcasting-type environmental information service aimed at a large audience [2][4].
(Before) The personalized healthcare service prompted us to develop predictive analytics technology, which requires the acquisition of data related to individual activity patterns [2,3]. Such data can be seen as pico-scale data that is different from the data to be used for the existing broadcasting-type environmental information service aimed at a large audience [2][4].
line 99: please define and explain what is "Kh"
Kh is a chosen kernel, an internel function to smooth the time-series data. A typical example of the Kernel includes the Gaussian kernel in the Kernel regression algorithm. The definition was provided in line 102 as requested.
Figures 6-9 and related text: there extremely high values of PM2.5 shown here, up to >8000 ug/m3. These are barely credible as realistic values: did the author double checked the data, e.g. where and in which environment they were detected?
Thank you for asking.
Several studies reported that indoor barbecue or indoor pan-frying contribute to the indoor emission of PM2.5 with such high level. According to our previous study conducted in Korea, June 2013, when there was indoor pan-frying of 100 g pork for only 9 min, the median (interquartile range, IQR) PM2.5 value was 4.5 (2.2–5.6) mg/m3 for no ventilation and 0.5 (0.1–1.3) mg/m3 with an active stove hood ventilation system over a 2 h sampling interval. Thus, the values of >8000 ug/m3 (8 mg/m3) were high and, at the same time, realistic values.
Reference: Lee S, Yu S, Kim S, Evaluation of Potential Average Daily Doses (ADDs) of PM2.5 for Homemakers Conducting Pan-Frying Inside Ordinary Homes under Four Ventilation Conditions, International Journal of Environmental Research and Public Health. 2017, 14, 78; doi:10.3390/ijerph14010078
I understand that this is mostly a paper presenting a new time series method, thus the physical meaning of the values may be overridden to some extent, but this seems to be really too much. I suggest to select another sample series to illustrate the results, e.g. the part after time 10.000 of the same series.
As suggested by the reviewer, three flat parts of the time-series data after time 10,000 were chosen to examine any performance differences. The corresponding results were provided at the bottom three parts of Table 1. The results were very similar for all three cases. Details on the chosen parts were described in line 144 to 147.
For reproducibility of the work, I strongly suggest to document the code and software that was used. At line 149 is mentioned R, but a detailed documentation of the calculation environments and packages used should be included, with proper acknowledgement and references.
Programming codes has been documented so far and will be updated if necessary, as suggested by the reviewer Detailed explanations on the computation environments and R packages (stats, zoo, Metrics) used, were provided in line 156 to 159. A related reference [18] was also added as requested.

Reviewer 2 Report
Regarding the manuscript entitled “Improved Interpolation and Anomaly Detection for Personal PM2.5 Measurement”. The manuscript is well written and presented, and the methodological approach is scientifically sound. Even though in my opinion the methodology is not yet complete, as it is acknowledged by the authors themselves, I believe that the study is still important, since it presents some first, very promising results of a new approach to overcome some very common difficulties in exposure studies. The main weaknesses of the study have to do with the non-automatic bandwidth selection and the visual inspection for the selection of the outliers. Since both of those points are mentioned by the authors and are set as future goals, my objections are not strong. What I would suggest, is that the authors need to provide some empirical approach for the determination of the threshold during the visual inspection because without it, it is very hard for the other researchers to use/replicate the methodology. Taking everything into account, I find the study suitable to be published after minor revisions.
Author Response
What I would suggest, is that the authors need to provide some empirical approach for the determination of the threshold during the visual inspection because without it, it is very hard for the other researchers to use/replicate the methodology.
In selecting a threshold, we took an empirical approach to decide the threshold. Some data that could be visually regarded as anomaly data were selected as reference data, and then these data were examined to see if they were detected as actual anomaly data. As we performed the experiments with varying thresholds, the values when these reference data were detected were selected as thresholds. The above comments were provided in line 225 to 228 and in line 266 to 267.

Round 2
Reviewer 1 Report
The authors replyied to my comments in the first round of review.